# RadEyeVideo: Enhancing general-domain Large Vision Language Model for chest X-ray analysis with video representations of eye gaze

## Abstract

Large Vision-Language Models (LVLMs) have demonstrated promising performance in chest X-ray (CXR) analysis. To enhance human-computer interaction, several studies have incorporated radiologists' eye gaze, typically through heatmaps or textual prompts. However, these methods often overlook the sequential order of eye movements, which could provide valuable insights by highlighting both the areas of interest and the order in which they are examined. In this work, we propose a novel approach called **RadEyeVideo** that integrates radiologists' eye-fixation data as a video sequence, capturing both the temporal and spatial dynamics of their gaze. The video, featuring a red gaze point overlaid on CXR images, emphasizes regions of focused attention during interpretation. We evaluate this method in CXR report generation and disease diagnosis using three general-domain, open-source LVLMs with a video input capabilities. When prompted with eye-gaze videos, model performance improves by up to 25.4% on Impression generation task and on average 7.9% for all tasks using scaled evaluation metrics. Our approach enhanced open-domain LVLM models, when combined with exemplar reports for in-context learning, outperform medical models as well as those specifically trained for CXR report generation on the benchmark dataset. This work highlights that domain expert's knowledge (eye-gaze information in this case), when effectively integrated with LVLMs, can significantly enhance general-domain models' capabilities in clinical tasks, pointing out a new effective approach of utilising LVLMs in healthcare and beyond.

## 1 Introduction

Large Vision-Language Models (LVLMs) have emerged as promising solutions for automating tasks in chest X-ray (CXR) analysis, including the generation of radiology reports, visual question answering, and error detection within medical reports (Bannur et al., 2024; Saab et al., 2024; Li et al., 2024; Wu et al., 2023; 2024a). These models offer the potential to streamline clinical workflows, providing radiologists with fast, automated insights that enhance decision-making and overall diagnostic efficiency.

However, despite these successes, the reliability of LVLMs in real-world clinical environments remains a challenge. A key limitation is the variability and accuracy of the outputs generated by these models (Xiao et al., 2024; AlSaad et al., 2024; Chen et al., 2024a; Wu et al., 2024b). A promising solution is to incorporate human expertise through human-computer interaction. Several studies have shown that integrating human input with AI models can significantly improve both accuracy and reliability, often exceeding the performance of radiologists and AI models when they work independently (Calisto et al., 2022; Patel et al., 2019).

One effective way to incorporate human expertise is through the use of radiologists' eye-tracking data collected while reviewing images. Studies have demonstrated that integrating eye-gaze information into AI models enhances diagnostic accuracy by providing insights into the areas radiologists focus on during image interpretation (Wang et al., 2022; Ma et al., 2023; Ji et al., 2023; Zhao et al., 2024b; Wang et al., 2024).

Recent work has expanded the use of radiologists' eye-tracking data in LVLMs for multimodal tasks like report generation and visual question answering (Kim et al., 2024a;b). These models typically integrate eye-gaze information through either simplified static textual prompts or heatmaps. However, these recent studies generally overlook the sequential order of eye movements, which could provide valuable additional context. The sequence in which a radiologist scans an image offers insight into how they prioritize different regions, potentially contributing to more nuanced interpretations and improving performance on downstream tasks.

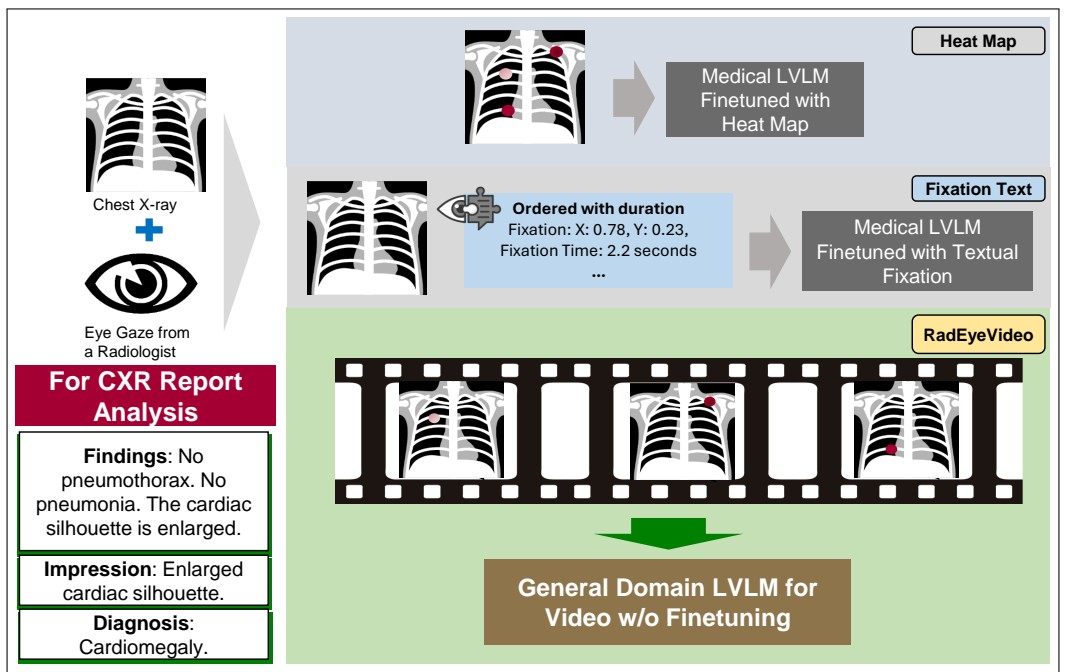

Figure 1: Comparison of the eye gaze prompting methods. **Heat Map:** Example of previous work that used the static simplified representation of eye gaze information laid over the chest X-ray image. **Fixation Text:** Example of previous work that implemented eye gaze information as textual prompt in the order of duration. **RadEyeVideo:** Our prompting method implements a video to capture a dynamic representation of eye gaze. To comply with the MIMIC-CXR data usage license, the CXR images are replaced with Pictogram and text reports are paraphrased in this Figure.

To address this gap, we propose a novel prompting approach, **RadEyeVideo**, that integrates eye-tracking data as a video, capturing both the temporal and spatial order of eye movements, also known as scan paths. This richer representation preserves the dynamic process of how radiologists navigate and prioritize different regions in an image. By incorporating this sequential flow, our approach offers deeper insights into the decision-making process, highlighting both the areas of interest and the order in which they are examined. Since radiologists often follow a structured approach during interpretation, capturing this sequence provides critical context for the model's understanding.

Figure 1 shows how our approach distinguishes itself from previous prompting methods for LVLMs in CXR analysis. The heatmap serves as a simplified snapshot of gaze movements, which fails to convey any information about their sequential order (Kim et al., 2024a). Although the textual prompt can contain some sequential temporal information regarding fixations, existing methods do not fully leverage this capability; instead, gaze data is typically organized by duration, limiting its representation of sequential order (Kim et al., 2024b). Although the textual format has the potential to convey a correct sequence, when ordered, it ultimately represents all spatial and temporal information in plain text. The X and Y coordinates, represented in relative width and height, offer only an indirect representation. In contrast, RadEyeVideo captures the dynamic nature of gaze patterns, providing a more comprehensive understanding of radiologist behavior during image interpretation.

This work aims to advance human-centered AI research in AI-assisted diagnostics through innovative integration of eye-tracking data. It establishes a generalizable prompting strategy with a video

that can be applied across various medical and non-medical fields. The contributions offer significant improvements in both the accuracy of report generation and diagnosis and the potential for broader human-AI collaboration. This multimodal human-in-the-loop (HITL) approach aims to advance human-centered AI research in medical image computing by effectively combining artificial and human intelligence. The proposed work will evaluate the impact of radiologists' expertise or perceptual cues in the form of eye gaze on model performance, aiming to enhance the accuracy and clinical relevance of AI-driven solutions for various medical image analysis tasks. Our approach offers several key contributions to the fields of AI-driven diagnostics and human-AI collaboration, summarized as follows:

- **RadEyeVideo - Dynamic Eye Gaze with Video for CXR Analysis**: We propose a novel method that integrates radiologists' eye gaze as video sequences for CXR report generation and diagnosis. This method leverages both spatial and temporal aspects of gaze patterns to significantly improve the performance of general-domain LVLMs in CXR report generation and diagnosis, achieving 25.4% on Impression generation and on average 7.9% performance boost on all tasks, outperforming task-specific medical models.

- **Comprehensive Benchmark of Eye-Gaze Prompting Methods**: Our study provides the first comprehensive evaluation of multiple eye-gaze integration techniques, including static heatmaps, fixation text, and dynamic video prompts. We demonstrate that RadEyeVideo consistently delivers superior results in terms of diagnostic accuracy and clinical relevance, setting a new standard for eye-gaze-based CXR analysis.

- **MIMIC-Eye-Video: Eye-Gaze Video Dataset for Chest X-rays** We present the MIMIC-Eye-Video dataset, the first to capture radiologists' eye-gaze as video during CXR interpretation. Although direct release is restricted by MIMIC-CXR policies, we provide code in the supplementary materials for researchers to recreate the dataset and advance medical AI by integrating expert gaze with CXR analysis.

## 2 RadEyeVideo - Dynamic Eye Gaze Video Prompting

We introduce RadEyeVideo, a dynamic eye-gaze video prompting technique, designed to incorporate radiologists' eye-gaze patterns into LVLMs for chest X-ray report generation and diagnosis. The key motivation behind this method is to enhance the interpretative capabilities of general-purpose LVLMs by leveraging human perceptual cues, such as eye-tracking data. The rationale for incorporating both spatial and temporal aspects of radiologists' gaze patterns is based on the understanding that expert radiologists do not analyze medical images in a static manner; rather, they focus on diagnostically relevant regions over time. Capturing these dynamics allows the model to gain insights from the way experts interact with medical images, thus guiding it toward better diagnostic decisions.

### 2.1 Why Eye-Gaze Data?

Radiologists demonstrate high proficiency in interpreting CXRs due to their ability to efficiently scan the image and focus on areas that are clinically significant. Eye-tracking data captures these visual search patterns and provides a rich source of information, reflecting the expert's decision-making process. RadEyeVideo translates these patterns into video-based prompts, which combine spatial attention (where radiologists look) with temporal sequencing (how long and when they look at certain regions), creating a multi-modal input that better aligns with clinical reasoning.

### 2.2 Constructing the Gaze-Based Video Prompt

Let $G = \{g_1, g_2, \ldots, g_n\}$ denote the sequence of gaze fixations for a CXR image, where each gaze fixation $g_i = (x_i, y_i, t_i)$ consists of the spatial coordinates $(x_i, y_i)$ and fixation duration $t_i$. The gaze radius size was fixed to 5 pixels.

To ensure we focus on the most significant gaze patterns, we filter gaze fixations based on the assumption that longer fixations are more diagnostically significant. We first calculate the average fixation duration $\bar{t}$ from the sequence. Then, we filter the gaze fixations to retain only those with a duration greater than the average duration:

$$G' = \{g_i \in G \mid t_i > \bar{t}\} \quad \text{where} \quad \bar{t} = \frac{1}{n} \sum_{i=1}^{n} t_i \tag{1}$$

We then construct the video by representing each fixation $g_i$ over a series of frames, with the number of frames $F_i$ proportional to the duration $t_i$:

$$F_{\text{total}} = \sum_{g_i \in G'} F_i = \sum_{g_i \in G'} (t_i \times 10) \quad \text{where} \quad F_i = t_i \times \text{fps} \tag{2}$$

where fps is the frame rate, set to 10 frames per second.

Each frame $v_j$ in the video sequence $V = \{v_1, v_2, \ldots, v_{F_{\text{total}}}\}$ represents the CXR image with a red dot at coordinates $(x_i, y_i)$, indicating the radiologist's gaze position. The duration of each fixation controls the number of frames in which the gaze remains in a given position.

## 2.3 INPUT REPRESENTATION

The input to the LVLM consists of two components: a textual prompt $T$ and the generated video sequence $V$. The textual prompt specifies the task (e.g., "Write a findings report on the given chest x-ray, including information about any abnormalities that you see."), while the video provides spatio-temporal information about the radiologist's eye movements.

To make the video suitable for LVLMs, which may require a fixed number of input frames (e.g., 16), we employ a uniform sampling strategy. Instead of using all frames, we evenly sample $k$ frames from the total video sequence $F_{\text{total}}$. The index of each sampled frame is given by:

$$V_{\text{sampled}} = \{v_j \mid j = 1, 2, \ldots, k\} \quad \text{where} \quad v_j = \left\lfloor \frac{j \cdot F_{\text{total}}}{k} \right\rfloor \quad \text{for} \quad j = 1, 2, \ldots, k \tag{3}$$

where $k$ is the number of frames to sample (typically 16), ensuring a balanced representation of the gaze data.

This sampling process, combined with the weighted number of frames based on duration, effectively captures important gaze patterns while reducing computational overhead. This approach maintains the temporal distribution and sequential order of gaze fixations, ensuring that key insights are preserved.

## 2.4 VIDEO PROMPTING FOR REPORT GENERATION

In clinical practice, radiologists' reports serve as crucial foundations for diagnostic and treatment decisions. CXR reports typically comprise two main sections: "Findings" and "Impressions." The "Findings" section meticulously details the radiologist's observations from the images, requiring keen observation skills and specialized knowledge. The "Impressions" section, on the other hand, provides a concise summary of the "Findings", offering clinicians a quick yet accurate diagnostic reference. In this case the report generation task can be split into two subtasks: **Findings generation** and **Impression generation**. Mathematically speaking,

$$Y_F = f_F(T, V, E) \tag{4}$$

$$Y_I = f_I(T, F, V, E) \tag{5}$$

where: $Y_F$ denotes the output of the Findings generation task, $Y_I$ denotes the output of the Impression summarization task, $T$ denotes the task instructions or prompts for each task, $V$ represents the sampled video prompts. To align AI-generated reports more closely with authentic clinical reporting styles, we provide three exemplar reports, $E$, as in-context learning. In the Impression geneartion task, $F$ denotes the Findings from original reports, which serves as an additional input for the Impression task.

## 2.5 VIDEO PROMPTING FOR DIAGNOSIS

Diagnosis is a critical component of chest radiograph (CXR) interpretation, where radiologists synthesize observed abnormalities to form a conclusive diagnostic assessment. This process demands not only acute visual perception but also extensive clinical knowledge and reasoning skills. In our study, we adapt our video-based eye-gaze prompting methodology to assist Large Vision-Language Models (LVLMs) in generating more accurate and clinically relevant diagnoses. The diagnostic task can be mathematically represented as:

$$Y_D = f_D(T, V, E) \tag{6}$$

where $Y_D$ denotes the output of the diagnostic task.

## 2.6 PROMPTS

Figure 2 provides a comprehensive overview of the various textual prompts used to enhance report generation and diagnosis for chest X-rays. The task prompt is a brief description of the task for findings, impression, and diagnosis respectively. Exemplars is the prompt for in-context learning of the CXR report writing style. For this, we provide three exemplar reports, only the text component. For the evaluation without any gaze information, this task prompt together with the exemplars is used, which will be called "NoEye" hereafter. Also, they serve as the base prompt for the eye gaze prompts. As described in Figure 1, which outlines the visual prompt inputs, the integration of textual prompts and raw images forms the "NoEye" and "Fixation Text" prompts. Similarly, the textual prompts are combined with eye gaze heatmap images for the "Heat Map" prompt and with eye gaze video sequences for the "RadEyeVideo" prompt.

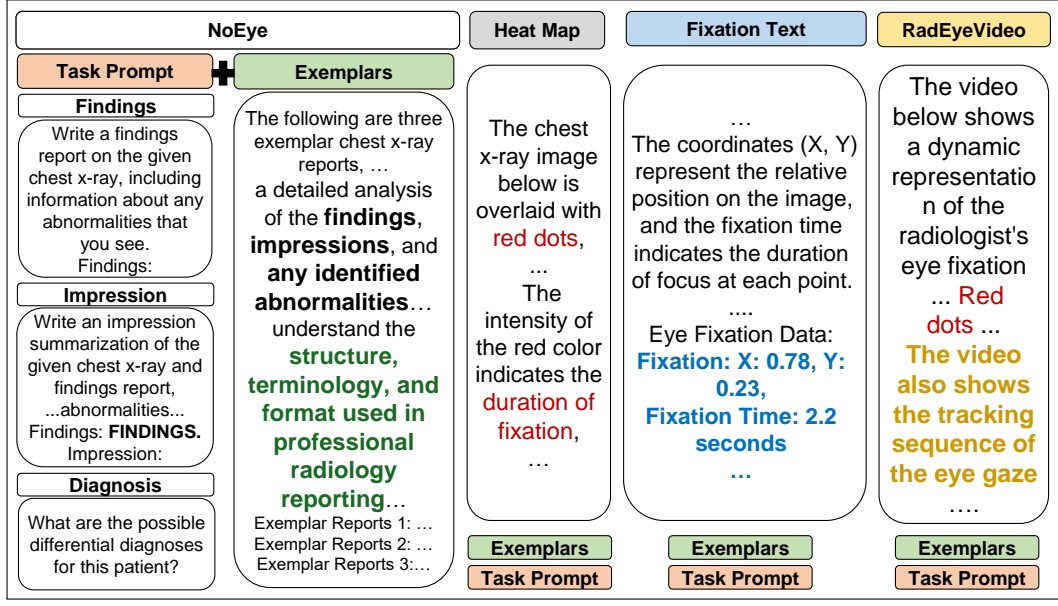

Figure 2: Eye gaze textual prompts. Texts are highlighted in different colors to emphasize the important aspect of each prompting method.

## 3 EXPERIMENT

### 3.1 EYE GAZE DATASET

For our study, we used the most commonly taken single-view chest X-ray, specifically the posterior-anterior (PA) view, commonly referred to as the frontal view, from the MIMIC-Eye dataset (Hsieh et al., 2023). This subset consists of a total of 2,298 CXR images that have been annotated with eye gaze data to capture radiologists' visual attention patterns while reviewing the images.

Table 1: Summary of images and tokens for findings and impressions

| Category | Alpha Set | Beta Set |
|---|---|---|
| Images w/ Diagnosis & Findings | 2,206 | 92 |
| Tokens (Findings) | $69.9 \pm 28.8$ | $86.7 \pm 35.1$ |
| Images w/ Diagnosis & Findings & Impressions | 1,963 | 62 |
| Tokens (Impressions) | $20.5 \pm 18.3$ | $27.8 \pm 22.6$ |

Since the original reports and images originate from the MIMIC-CXR dataset, we examined the overlap with its training and test splits (Johnson et al., 2019). Our analysis revealed an imbalanced distribution within the MIMIC-Eye subset, with the validation and test split containing only 92 images. This limited sample size is insufficient for robust evaluation purposes. To address this limitation, we chose to utilize both the training and test splits, which we will refer to as **alpha** and **beta** set hereafter for clarity and simplicity, as our evaluation dataset while ensuring that we report the results separately. This strategy enables us to trace any potential contamination issues that may arise from including training images in the evaluation process.

In the following table (Table 1), we provide a summary of the dataset statistics, including the number of images with findings, the average number of tokens in the findings and impressions sections, and how these metrics differ between the alpha and beta splits. The token statistics were measured based on the LLaVA-OneVision's tokenizer (Li et al., 2024).

## 3.2 MODELS

Table 2: Model descriptions. Training features are abbreviated as follows: Findings (F), Impression (I), and Diagnosis (D). Models in bold are trained with the MIMIC-CXR dataset.

| Model Name | Size | Backbone LLM | Report Trained | | | Supported Modalities | | |
|---|---|---|---|---|---|---|---|---|
| | | | F | I | D | Image | Text | Video |
| **CXRMate**(Nicolson et al., 2023) | 0.1B | - | ✓ | ✓ | ✗ | ✓ | ✗ | ✗ |
| **CheXagent**(Chen et al., 2024b) | 8B | Mistral 7B | ✓ | ✓ | ✓ | ✓ | ✓ | ✗ |
| **CXR-LLaVA**(Lee et al., 2023) | 8B | LLaMA2 7B | ✓ | ✗ | ✓ | ✓ | ✓ | ✗ |
| LLaVA-Med(Li et al., 2024) | 8B | Mistral 7B | - | - | - | ✓ | ✓ | ✗ |
| LongVA(Zhang et al., 2024) | 8B | Qwen2 7B | - | - | - | ✓ | ✓ | ✓ |
| VideoLLaMA2(Cheng et al., 2024) | 8B | Mistral 7B | - | - | - | ✓ | ✓ | ✓ |
| LLaVA-OneVision(Li et al., 2024) | 8B | Qwen2 7B | - | - | - | ✓ | ✓ | ✓ |

In this study, we selected a range of models to investigate various prompting methods for integrating eye gaze information into the CXR report generation process. Specifically, we focused on LongVA, VideoLLaMA2, and LLaVA-OneVision, which are the latest LVLMs that can handle videos (Zhang et al., 2024; Cheng et al., 2024; Li et al., 2024).

For baseline comparisons, we included CXR LVLMs such as CXR-LLaVA and CheXagent (Lee et al., 2023; Chen et al., 2024b). Both of these models have been trained with the entire MIMIC-CXR training split which is about 200,000 CXR images and reports for diverse tasks, including report generation.

Additionally, LLaVA-Med based on Mistral is used as a general medical LVLM (Li et al., 2024; Liu et al., 2023). It has not been trained on the MIMIC-CXR dataset, but it has been trained on medical images and captions derived from biomedical articles, specifically from PubMed.

Lastly, we included CXRMate, which represents the state-of-the-art CXR report generation model which accepts only images as the input (Nicolson et al., 2023). Notably, all models, except CXR-Mate, are built upon a 7B backbone large language model, highlighting the varying capacities and training methodologies across the selected models.

## 3.3 EVALUATION

In addition to using eye-tracking data as video sequences, we explore other methods of integrating eye-gaze information into LVLMs for CXR report analysis, including heatmaps and textual prompts. We systematically evaluate these methods to determine which most effectively enhances report generation, focusing on accuracy, completeness, and clinical relevance.

In implementing heatmaps, we overlay red dots on the CXR images to visualize spatial patterns of eye gaze information. These dots indicate the areas of focus, with darker dots representing longer gaze durations, thereby incorporating a temporal dimension to the data. For our textual prompt approach, we maintain the original sequence of gaze information rather than organizing it by duration. This approach still has a limitation, indirect representation of the spatial representation. The relative position of the gaze in the text has to be used to infer the gaze point on the image, and LVLMs capability of doing this remains unexplored. Similar to the video prompting method, we filter the gaze data to retain only those fixations that exceed the average duration, ensuring that our analyses focus on significant eye-gaze interactions.

### 3.3.1 EVALUATION HYPERPARAMETERS

For evaluation, we implemented a zero-shot approach for both sections, a batch size of 1, and a temperature parameter of 0. A temperature of 0 was chosen to minimize the randomness in the generated text produced by the model. The maximum length of the model's responses for each task was determined based on the mean token length of the ground truth response: 96 for the findings section and 32 for the impression section. For the diagnosis, we wanted the model to generate a list of possible diagnosis, so the maximum length was set as 192. This setup ensured consistent and efficient experimentation across both tasks.

### 3.3.2 EVALUATION METRIC

We evaluated the generated reports using a combination of general lexical metrics and radiology-specific metrics. For general assessment, we employed ROUGE and BERTScore (Lin, 2004; Zhang et al., 2019). In terms of radiology-specific evaluation, we utilized CheXbert (micro F1 for top 5 abnormalities), RadGraph, and RaTEScore (Smit et al., 2020; Jain et al., 2021; Zhao et al., 2024a). For the diagnosis, as the task focuses on the generation of possible diagnosis, rather than focusing on the top 5 abnormalities, we used CheXbert micro classification F1 for all abnormalities as the metric (Smit et al., 2020).

To compare the performance of our models with respect to both general lexical and radiology-specific metrics, we introduce a scaling based on CheXagent, a medical LVLM that is trained for both report generation and diagnosis and known to perform the best among the LVLMs for these two tasks (Chen et al., 2024b). This scaling ensures that all metrics are directly comparable and highlights relative improvements (or declines) across models, regardless of the absolute values.

Let $S_{m,i}$ represent the raw score for metric $m$ on model $i$, and let $S_{m,CheXagent}$ represent the score of the LLaVA-Med for the same metric $m$. The scaled score $\hat{S}_{m,i}$ for each model $i$ on metric $m$ and the average of the scaled scores across all metrics $M$ for the the overall performance score, $\hat{S}_i$, are then calculated as follows:

$$\hat{S}_{m,i} = \frac{S_{m,i}}{S_{m,CheXagent}} \times 100, \qquad \hat{S}_i = \frac{1}{|M|} \sum_{m \in M} \hat{S}_{m,i} \tag{7}$$

This average score provides a holistic measure of model performance on the dataset, combining general lexical metrics and radiology-specific metrics into a single representative score. This approach enables straightforward comparison across models and highlights the effectiveness of our novel video prompting in improving performance.

Table 3: Eye gaze evaluation results. All scores are averages of scaled metrics based on CheXagent. NoEye prompting scores are reported with model names. Alpha - the original MIMIC-CXR training split; Beta - the validation and test splits. Bold - best scores (excluding scores from MIMIC-CXR trained models); parentheses - performance improvements. *CXRMate does not support textual prompts for in-context learning; thus, its default performance is reported.

| Methods | Findings | | Impression | | Diagnosis | | Overall |
| | Alpha | Beta | Alpha | Beta | Alpha | Beta | |
|---|---|---|---|---|---|---|---|
| CXRMate* | 513.7 | 377.5 | 74.0 | 66.3 | - | - | - |
| CheXagent | 100.0 | 100.0 | 100.0 | 100.0 | 100.0 | 100.0 | 100.0 |
| CXR-LLaVA | 369.0 | 257.9 | 70.4 | 67.2 | 118.2 | 142.4 | 170.8 |
| LLaVA-Med | 236.7 | 215.9 | 51.8 | 55.1 | 117.7 | 142.8 | 136.7 |
| LongVA | 213.2 | 205.2 | 49.2 | 53.8 | 155.6 | 188.0 | 144.2 |
| w/ Heat Map | 216.4 | 193.2 | 53.0 | 57.2 | 153.9 | 200.2 | 145.7 |
| w/ Text | 232.5 | 197.6 | 56.5 | 56.5 | 162.5 | 209.8 | 152.6 |
| w/ RadEyeVideo (Ours) | 237.3 | 220.9 | 52.3 | 53.9 | 159.7 | 211.8 | 156.0 |
| | **(+24.1)** | **(+15.7)** | **(+3.1)** | **(+0.1)** | **(+4.1)** | **(+23.8)** | **(+11.8)** |
| VideoLLaMA2 | 226.5 | 203.4 | 59.1 | 75.5 | 163.5 | 207.2 | 155.9 |
| w/ Heat Map | 238.3 | 199.8 | 69.1 | 80.8 | 155.6 | 210.1 | 159.0 |
| w/ Text | 251.1 | 210.3 | 66.6 | 80.2 | 165.9 | 208.1 | 163.7 |
| w/ RadEyeVideo (Ours) | 244.1 | 205.7 | 70.0 | 80.9 | 150.6 | 188.0 | 156.5 |
| | (+17.6) | (+2.3) | (+10.9) | (+5.4) | (-12.9) | (-19.2) | (+0.6) |
| LLaVA-OneVision | 254.2 | 231.5 | 53.4 | 56.4 | 170.0 | 219.3 | 164.1 |
| w/ Heat Map | 234.4 | 203.3 | **71.5** | 79.6 | 172.7 | 223.7 | 164.2 |
| w/ Text | 230.1 | 208.4 | 68.6 | 77.7 | 153.4 | 180.8 | 153.2 |
| w/ RadEyeVideo (Ours) | **259.6** | **235.3** | 70.4 | **81.8** | **178.3** | **226.1** | **175.3** |
| | **(+5.4)** | **(+3.8)** | **(+17.0)** | **(+25.4)** | **(+8.3)** | **(+6.8)** | **(+11.2)** |

# 4 RESULTS AND DISCUSSION

The evaluation results in Table 3 offer a comprehensive overview of performance across two sections: Findings and Impression, evaluated for both the alpha and beta splits. Overall, general domain models when eye gaze information is provided regardless of the method consistently outperform the NoEye prompting method, achieving higher average scores except for the LLaVA-OneVision model with the fixation text prompting.

**RadEyeVideo Prompting as the Optimal Method**
Our proposed method, RadEyeVideo, emerges as the most robust and effective method. Across all three general domain models—LongVA, VideoLLaMA2, and LLaVA-OneVision—video prompting consistently outperforms the NoEye method, demonstrating reliable performance improvements. The only exception is VideoLLaMA2 for the diagnosis task where RadEyeVideo only showed a decrease in performance. While heat map increased 1.6% and fixation text increased 1.8% on average, our RadEyeVideo increased 7.9% on average. Specifically, LongVA, VideoLLaMA2, and LLaVA-OneVision improve their average performance by 11.8%, 0.6%, and 11.2%, respectively. Remarkably, the LLaVA-OneVision model with RadEyeVideo surpasses both CheXagent and CXR-LLaVA models, which were specifically trained on the MIMIC-CXR dataset for report generation and diagnosis. This finding underscores the potential of video-based eye-gaze prompting to bridge the performance gap between general-purpose models and task-specific models in the medical imaging domain.

**Limitations of Heatmap and Fixation Text Prompts**
While RadEyeVideo shows strong results, both heatmap prompting and fixation text prompt methods do not always improve performance. While fixation text prompts show mixed results-VideoLLaMA2 always gaining performance, heatmap-based prompting universally underperforms when compared to the baseline across all models for the generation of the Findings section. The LLaVA-OneVision model suffers from a performance drop in both splits when using heatmaps. Also, the LongVA model shows a consistent drop in performance when utilizing heatmaps for the

beta split and the VideoLLaMA2 model suffers a similar degradation. These two models also show a decrease in diagnosis performance for the alpha split.

The poor performance of heatmaps is likely due to their failure to capture the sequential and temporal dynamics of eye-gaze, which are essential for understanding medical analysis in chest X-rays. This supports our hypothesis that the temporal order of a radiologist's gaze is critical for enhancing contextual awareness and clinical relevance.

Fixation text prompts, although they preserve the temporal sequence of eye-gaze, suffer from other limitations. By attempting to encode both spatial and temporal information in a text format, these prompts often become overly lengthy and complex, particularly when the model is asked to generate a longer response, the Findings and Diagnosis. As a result, LongVA and LLaVA-OneVision both performed worse in the Findings section, and LLaVA-OneVision performed worse in Diagnosis.

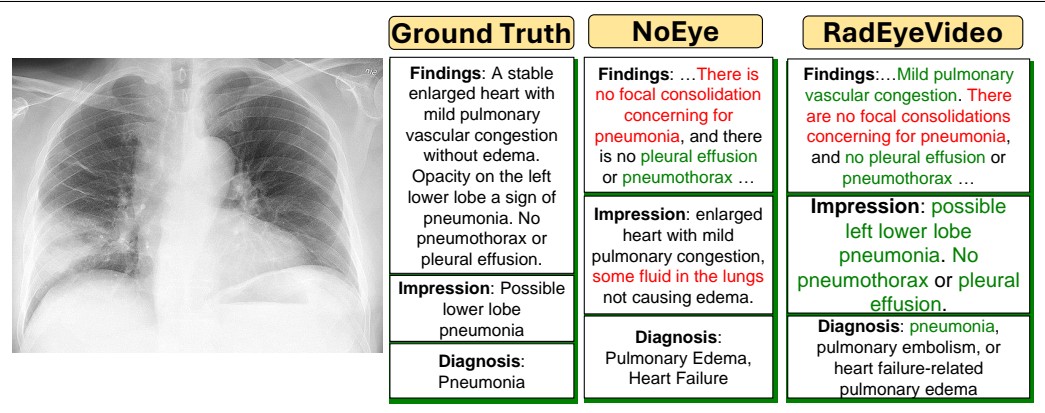

Figure 3: Sample response from LLaVA-OneVision. **Green** - correct statements about the CXR; **Red** - incorrect statements. To comply with the MIMIC-CXR data usage license, the CXR image is substituted with a Wikimedia image depicting the same disease, and the text report is paraphrased.

**Singificant Improvement on the Impression Section**
The Impression section showed the most significant performance boost, particularly with video prompting. This is likely due to the added context from the Findings section, which allowed the models to make more informed and accurate predictions. Among all models, LLaVA-OneVision demonstrated the largest improvement, with video prompting leading to a 17.0% performance increase on the alpha split and an impressive 25.4% gain on the beta split. These results suggest that video-based prompting, which preserves both spatial and temporal aspects of eye-gaze information, offers richer guidance to the model, leading to more precise and contextually aware outputs. Figure 3 exemplifies this improvement in the Impression section, showing a significant change in the generated text, correctly finding pneumonia. While the Findings section showed slight improvements with more medical concepts correctly stated, errors like the misdiagnosis of pneumonia remain, indicating room for further improvement. Figure 3 also shows that Diagnosis is also improved with RadEyeVideo as the model correctly predicted pneumonia.

Moreover, both VideoLLaMA2 and LLaVA-OneVision not only outperformed medical models such as LLaVA-Med and CXR-LLaVA, but they also surpassed the performance of the state-of-the-art model CXRMate in the Impression section for the test split. This is remarkable, given that CXRMate was specifically designed for chest X-ray report generation and represents the current benchmark in the field.

## 4.1 ABLATION STUDY

The ablation study was conducted to further validate the design of RadEyeVideo, focusing on three critical factors: frame numbers, gaze point sampling, and in-context learning.

**Frame Numbers** Figure 4 highlights the impact of varying the number of frames on model performance. Our findings indicate that utilizing 16 frames achieves the highest average score of 161.8%.

Increasing the frames to 32 results in a slight decline of 2.9%, while 64 and 128 frames lead to further decreases of 9.2% and 12.6%, respectively. This trend indicates that longer video sequences may hinder LVLM performance due to increased input complexity, reinforcing our decision to use 16 frames.

**Gaze Point Filtering** The second aspect examined was the impact of gaze point filtering by duration, as detailed in Section 2. Figure 4 shows that significant gaze point sampling has a minimal effect on model performance. Specifically, LongVA and LLaVA-OneVision improved by 1.4% and 1.7%, respectively, while VideoLLaMA2 experienced a slight decline of 1.7%. Since two of the three models benefited from this filtering, we chose to implement significant gaze filtering.

**In-Context Learning with Exemplar Reports** Lastly, we examined the effect of in-context learning on various models. Models that have been trained for the task, such as CheXagent and CXR-LLaVA, experienced significant declines in performance, with average losses of -130.6% and -40.5%, indicating that these models are not robust to in-context learning and may have been overfit to the task. Other models like LLaVA-OneVision, LongVA, VideoLLaMA2, and LLaVA-Med showed varying improvements of 35.4%, 50.3%, 65.7%, and 81.9%, respectively. This ablation study suggests that models not trained on MIMIC-CXR data benefit more from in-context learning.

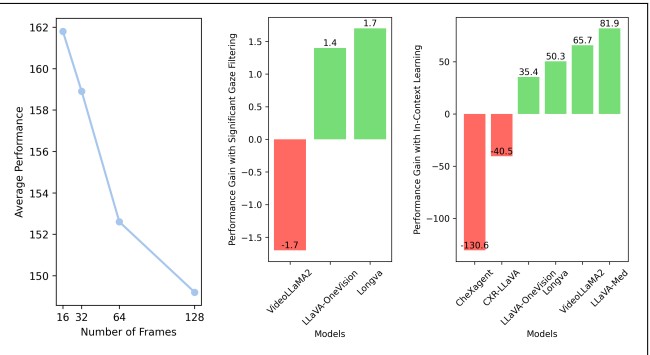

Figure 4: Results for the ablation study of the experiment design. **Left:** frame numbers, **middle:** significant gaze filtering, **right:** exemplar reports in-Context learning.

## 5 CONCLUSION AND FUTURE WORK

In this work, we introduced RadEyeVideo, a novel video prompting approach, into general domain LVLMs, enhancing their performance in chest X-ray report generation and diagnosis. Our approach effectively captures the spatial and temporal dynamics of eye gaze, outperforming existing eye-gaze prompting techniques and even the task specific medical LVLMs. This method demonstrates significant potential for bridging the gap between general-purpose and task-specific models in medical imaging. Future work will explore extending video-based prompting to tasks like visual question answering and anatomical structure detection. Additionally, applying this method to other medical imaging areas, such as CT and MRI scans, holds promise for improving accuracy and clinical relevance across various domains in healthcare.

### LIMITATION

This study's evaluation was limited by the small dataset size, due to the difficulty of obtaining radiologists' eye-tracking data. Furthermore, the MIMIC-CXR dataset may not fully capture the diversity of real-world medical imaging. Future work should focus on larger and more diverse datasets to better assess the method's generalizability across different imaging modalities.

### ETHICAL STATEMENT

This research adhered to the data usage agreements of the MIMIC-EYE dataset and maintained strict compliance with privacy regulations.

### REPRODUCIBILITY STATEMENT

All models and datasets in this study are publicly available. We included our code to process the datasets in the supplementary material.

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

# A  APPENDIX

## A.1  COMPUTATIONAL EFFICIENCY OF RADEYEVIDEO

The incorporation of eye gaze data as video, rather than as a heatmap, results in only a marginal increase in GPU memory usage—approximately 2GB, rising from 20,076MB (eye gaze heatmap) to 21,954MB (eye gaze video) when using the LLaVA-OneVision model. This corresponds to a modest 10% increase in GPU memory consumption.

Notably, this increase is minimal when compared to the significantly higher computational demands of training specialized medical LVLMs, which often require at least double the computational resources. The efficiency of our approach underscores its practicality for deployment, even in resource-constrained environments, while achieving enhanced diagnostic performance.

## A.2  RESPONSE EXAMPLES

We demonstrate two examples of the generated response of the LLaVA-OneVision for all te prompting methods that we explored in our study: pneumonia 5 and fracture 6. In both examples, inclusion of eye gaze in any formats, the model was able to produce some medically accurate response. However, the heat map and fixation text prompting generated wrong information. Our RadEyeVideo prompting had the best Findings and Impression, and even a correct diagnosis for the pneumonia case.

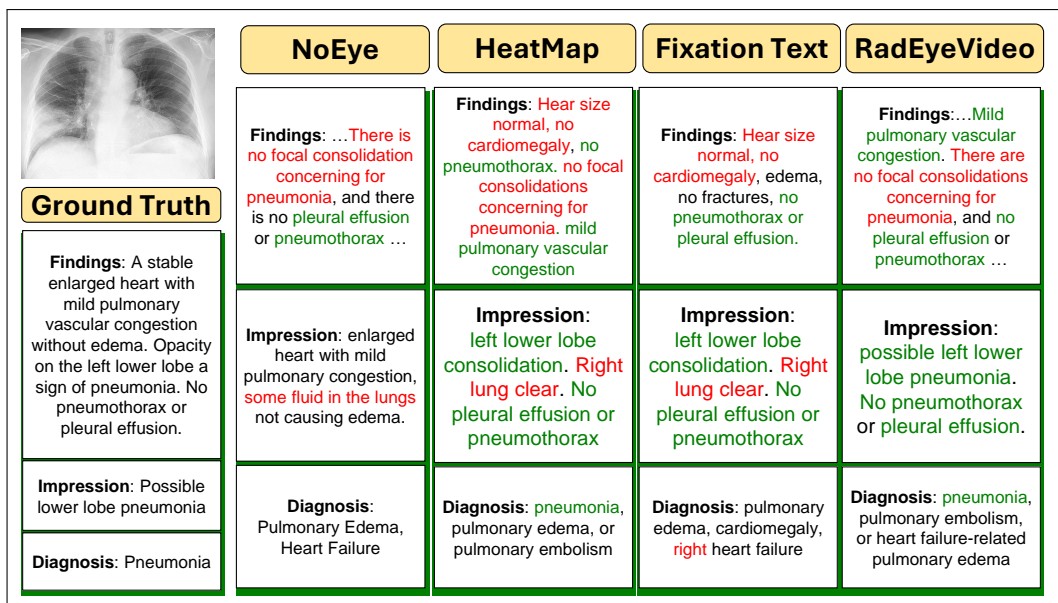

Figure 5: Pneumonia response by LLaVA-OneVision for all prompting methods. Green shows the correct statement about the CXR, and the red shows the incorrect statement. To comply with the MIMIC-CXR data usage license, the CXR image is replaced with an image from Wikimedia with the same disease and the text report is paraphrased in this Figure.

## A.3  EVALUATION METRICS EXPLANATION

**ROUGE Lin (2004)** ROUGE is a widely used lexical similarity metric primarily based on recall of the token matches. For this study, we use ROUGE-L, which measures the longest common subsequence and an F-score to offer a more comprehensive performance assessment.

**BERTScore Zhang et al. (2019)** BERTScore goes beyond exact token matching by comparing the similarity between tokens in candidate and reference sentences using contextual embeddings of the BERT model, offering a more nuanced evaluation of text similarity.

Figure 6: Fracture response by LLaVA-OneVision for all prompting methods. Green shows the correct statement about the CXR, and the red shows the incorrect statement. To comply with the MIMIC-CXR data usage license, the CXR image is replaced with an image from Wikimedia with the same disease and the text report is paraphrased in this Figure.

**CheXbert Smit et al. (2020)** The CheXbert model is a BERT model which has been trained on large-scale chest radiograph data to extract abnormalities. We used micro-F1 score for our study: top-5 for the report generation, and all abnormalities for disease diagnosis.

**RadGraph Jain et al. (2021)** RadGraph extracts clinical entities (such as anatomy and observations) and their relationships from radiology reports, organizing them into graph structures. The RadGraph F1 score calculates entity and relation overlap separately, averaging the results. Entity matches are based on identical text spans and types, while relation matches require agreement in both endpoint entities and relation types.

**RaTEScore Zhao et al. (2024a)** RaTEScore (Radiological Report Text Evaluation) is the latest metric employed in our study, specifically designed to assess the quality of generated medical reports. It evaluates the correct identification of key medical entities, including diagnostic outcomes and anatomical references, using a NER model. RaTEScore is robust at handling complex medical synonyms and is sensitive to negation, making it more aligned with human judgments compared to other metrics.

## A.4 REPORT GENERATION PERFORMANCE

We report the performance of report generation in detail with the raw values for each metric.

Table 4: Report generation performance with ROUGE (ROUGE-L). All scores are raw scores reported. NoEye prompting scores are reported with the model name. *CXRMate does not support textual prompts for in-context learning, so its vanilla performance is reported instead.

| Methods | Findings | | Impression | | Overall |
|---|---|---|---|---|---|
| | Alpha | Beta | Alpha | Beta | |
| CXRMate*(Nicolson et al., 2023) | **0.304** | **0.256** | 0.370 | 0.229 | **0.290** |
| CheXagent(Chen et al., 2024b) | 0.081 | 0.083 | **0.531** | **0.381** | 0.269 |
| CXR-LLaVA(Lee et al., 2023) | 0.220 | 0.184 | 0.355 | 0.198 | 0.239 |
| LLaVA-Med(Li et al., 2024) | 0.206 | 0.199 | 0.131 | 0.123 | 0.165 |
| LongVA(Zhang et al., 2024) | 0.185 | 0.177 | 0.117 | 0.115 | 0.148 |
| w/ Heat Map | 0.181 | 0.181 | 0.172 | 0.146 | 0.170 |
| w/ Text | 0.196 | 0.192 | 0.213 | 0.149 | 0.188 |
| w/ RadEyeVideo (Ours) | 0.192 | 0.182 | 0.165 | 0.135 | 0.168 |
| VideoLLaMA2(Cheng et al., 2024) | 0.179 | 0.168 | 0.101 | 0.119 | 0.142 |
| w/ Heat Map | 0.205 | 0.182 | 0.237 | 0.194 | 0.204 |
| w/ Text | 0.200 | 0.180 | 0.189 | 0.176 | 0.186 |
| w/ RadEyeVideo (Ours) | 0.216 | 0.197 | 0.251 | 0.188 | 0.213 |
| LLaVA-OneVision(Li et al., 2024) | 0.189 | 0.185 | 0.134 | 0.121 | 0.157 |
| w/ Heat Map | 0.190 | 0.180 | 0.306 | 0.245 | 0.230 |
| w/ Text | 0.186 | 0.184 | 0.276 | 0.221 | 0.217 |
| w/ RadEyeVideo (Ours) | 0.192 | 0.181 | 0.296 | 0.252 | 0.230 |

Table 5: Report generation performance with BERTScore. All scores are raw scores reported. NoEye prompting scores are reported with the model name. *CXRMate does not support textual prompts for in-context learning, so its vanilla performance is reported instead.

| Methods | Findings | | Impression | | Overall |
|---|---|---|---|---|---|
| | Alpha | Beta | Alpha | Beta | |
| CXRMate*(Nicolson et al., 2023) | **0.888** | **0.874** | 0.896 | 0.876 | **0.884** |
| CheXagent(Chen et al., 2024b) | 0.848 | 0.845 | **0.922** | **0.897** | 0.878 |
| CXR-LLaVA(Lee et al., 2023) | 0.873 | 0.862 | 0.897 | 0.868 | 0.875 |
| LLaVA-Med(Li et al., 2024) | 0.865 | 0.862 | 0.855 | 0.852 | 0.858 |
| LongVA(Zhang et al., 2024) | 0.865 | 0.860 | 0.853 | 0.849 | 0.857 |
| w/ Heat Map | 0.860 | 0.858 | 0.857 | 0.853 | 0.857 |
| w/ Text | 0.865 | 0.859 | 0.864 | 0.854 | 0.860 |
| w/ RadEyeVideo (Ours) | 0.865 | 0.859 | 0.855 | 0.850 | 0.857 |
| VideoLLaMA2(Cheng et al., 2024) | 0.858 | 0.853 | 0.849 | 0.853 | 0.853 |
| w/ Heat Map | 0.862 | 0.855 | 0.872 | 0.866 | 0.864 |
| w/ Text | 0.861 | 0.854 | 0.865 | 0.863 | 0.861 |
| w/ RadEyeVideo (Ours) | 0.864 | 0.857 | 0.875 | 0.865 | 0.865 |
| LLaVA-OneVision(Li et al., 2024) | 0.862 | 0.857 | 0.854 | 0.849 | 0.856 |
| w/ Heat Map | 0.863 | 0.857 | 0.881 | 0.869 | 0.868 |
| w/ Text | 0.864 | 0.858 | 0.877 | 0.866 | 0.866 |
| w/ RadEyeVideo (Ours) | 0.864 | 0.858 | 0.881 | 0.872 | 0.869 |

## A.5 DIAGNOSIS PERFORMANCE

We report the performance of disease diagnosis in detail with the raw values for each abnormalities. The abnormalities covered in CheXbert are as follows: no finding, enlarged cardiomediastinum, cardiomegaly, Lung Lesion, Lung opacity, edema, consolidation, pneumonia, atelectasis, pneumothorax, pleural effusion, pleural other, fracture, support devices.

Table 6: Report generation performance with CheXbert (micro F1 for top 5 abnormalities). All scores are raw scores reported. NoEye prompting scores are reported with the model name. *CXR-Mate does not support textual prompts for in-context learning, so its vanilla performance is reported instead.

| Methods | Findings | | Impression | | Overall |
|---|---|---|---|---|---|
| | Alpha | Beta | Alpha | Beta | |
| CXRMate*(Nicolson et al., 2023) | **0.556** | **0.537** | 0.465 | 0.389 | **0.487** |
| CheXagent(Chen et al., 2024b) | 0.059 | 0.085 | **0.680** | 0.561 | 0.346 |
| CXR-LLaVA(Lee et al., 2023) | 0.401 | 0.372 | 0.479 | 0.431 | 0.421 |
| LLaVA-Med(Li et al., 2024) | 0.142 | 0.197 | 0.416 | 0.306 | 0.265 |
| LongVA(Zhang et al., 2024) | 0.096 | 0.180 | 0.389 | 0.322 | 0.246 |
| w/ Heat Map | 0.087 | 0.097 | 0.333 | 0.289 | 0.202 |
| w/ Text | 0.091 | 0.087 | 0.370 | 0.282 | 0.207 |
| w/ RadEyeVideo (Ours) | 0.112 | 0.198 | 0.328 | 0.247 | 0.221 |
| VideoLLaMA2(Cheng et al., 2024) | 0.138 | 0.183 | 0.616 | 0.675 | 0.403 |
| w/ Heat Map | 0.128 | 0.157 | 0.665 | 0.667 | 0.404 |
| w/ Text | 0.145 | 0.193 | 0.664 | **0.680** | 0.420 |
| w/ RadEyeVideo (Ours) | 0.139 | 0.174 | 0.673 | 0.676 | 0.416 |
| LLaVA-OneVision(Li et al., 2024) | 0.206 | 0.240 | 0.462 | 0.358 | 0.317 |
| w/ Heat Map | 0.104 | 0.138 | 0.598 | 0.550 | 0.347 |
| w/ Text | 0.145 | 0.182 | 0.574 | 0.585 | 0.371 |
| w/ RadEyeVideo (Ours) | 0.170 | 0.278 | 0.592 | 0.567 | 0.402 |

Table 7: Report generation performance with RadGraph. All scores are raw scores reported. NoEye prompting scores are reported with the model name. *CXRMate does not support textual prompts for in-context learning, so its vanilla performance is reported instead.

| Methods | Findings | | Impression | | Overall |
|---|---|---|---|---|---|
| | Alpha | Beta | Alpha | Beta | |
| CXRMate*(Nicolson et al., 2023) | **0.244** | **0.186** | 0.269 | 0.140 | 0.210 |
| CheXagent(Chen et al., 2024b) | 0.025 | 0.027 | **0.468** | **0.345** | **0.216** |
| CXR-LLaVA(Lee et al., 2023) | 0.161 | 0.106 | 0.296 | 0.148 | 0.178 |
| LLaVA-Med(Li et al., 2024) | 0.117 | 0.102 | 0.115 | 0.118 | 0.113 |
| LongVA(Zhang et al., 2024) | 0.109 | 0.098 | 0.089 | 0.090 | 0.097 |
| w/ Heat Map | 0.121 | 0.107 | 0.122 | 0.111 | 0.115 |
| w/ Text | 0.132 | 0.112 | 0.134 | 0.104 | 0.121 |
| w/ RadEyeVideo (Ours) | 0.131 | 0.112 | 0.117 | 0.091 | 0.113 |
| VideoLLaMA2(Cheng et al., 2024) | 0.111 | 0.097 | 0.126 | 0.175 | 0.127 |
| w/ Heat Map | 0.122 | 0.096 | 0.166 | 0.188 | 0.143 |
| w/ Text | 0.132 | 0.100 | 0.164 | 0.188 | 0.146 |
| w/ RadEyeVideo (Ours) | 0.121 | 0.094 | 0.167 | 0.184 | 0.142 |
| LLaVA-OneVision(Li et al., 2024) | 0.113 | 0.113 | 0.108 | 0.104 | 0.110 |
| w/ Heat Map | 0.132 | 0.110 | 0.203 | 0.195 | 0.160 |
| w/ Text | 0.111 | 0.100 | 0.186 | 0.170 | 0.142 |
| w/ RadEyeVideo (Ours) | 0.133 | 0.106 | 0.191 | 0.205 | 0.159 |

Table 8: Report generation performance with RaTEScore. All scores are raw scores reported. NoEye prompting scores are reported with the model name. *CXRMate does not support textual prompts for in-context learning, so its vanilla performance is reported instead.

| Methods | Findings | | Impression | | Overall |
|---|---|---|---|---|---|
| | Alpha | Beta | Alpha | Beta | |
| CXRMate*(Nicolson et al., 2023) | **0.579** | **0.513** | 0.512 | 0.374 | **0.495** |
| CheXagent(Chen et al., 2024b) | 0.354 | 0.339 | **0.663** | **0.589** | 0.486 |
| CXR-LLaVA(Lee et al., 2023) | 0.506 | 0.444 | 0.361 | 0.396 | 0.427 |
| LLaVA-Med(Li et al., 2024) | 0.421 | 0.421 | 0.371 | 0.350 | 0.391 |
| LongVA(Zhang et al., 2024) | 0.479 | 0.447 | 0.367 | 0.357 | 0.413 |
|   w/ Heat Map | 0.448 | 0.445 | 0.428 | 0.406 | 0.432 |
|   w/ Text | 0.478 | 0.452 | 0.436 | 0.399 | 0.441 |
|   w/ RadEyeVideo (Ours) | 0.466 | 0.454 | 0.426 | 0.406 | 0.438 |
| VideoLLaMA2(Cheng et al., 2024) | 0.470 | 0.466 | 0.444 | 0.471 | 0.463 |
|   w/ Heat Map | 0.468 | 0.456 | 0.485 | 0.490 | 0.475 |
|   w/ Text | 0.469 | 0.453 | 0.470 | 0.488 | 0.470 |
|   w/ RadEyeVideo (Ours) | 0.460 | 0.447 | 0.487 | 0.499 | 0.473 |
| LLaVA-OneVision(Li et al., 2024) | 0.470 | 0.441 | 0.386 | 0.363 | 0.415 |
|   w/ Heat Map | 0.470 | 0.428 | 0.484 | 0.485 | 0.467 |
|   w/ Text | 0.461 | 0.443 | 0.474 | 0.472 | 0.463 |
|   w/ RadEyeVideo (Ours) | 0.493 | 0.457 | 0.483 | 0.503 | 0.484 |

Table 9: Alpha Split Diagnosis performance with CheXbert (F1 score for all abnormalities). All scores are raw scores reported. NoEye prompting scores are reported with the model name.

| Methods | No Finding | Enlarged Cardiomediastinum | Cardiomegaly | Lung Lesion | Lung Opacity | Edema | Consolidation |
|---|---|---|---|---|---|---|---|
| CheXagent(Chen et al., 2024b) | 0.000 | 0.000 | 0.000 | 0.000 | 0.025 | **0.338** | **0.131** |
| CXR-LLaVA(Lee et al., 2023) | 0.000 | 0.000 | 0.000 | 0.000 | 0.012 | 0.209 | 0.000 |
| LLaVA-Med(Li et al., 2024) | 0.000 | 0.000 | 0.000 | 0.000 | 0.000 | 0.209 | 0.000 |
| LongVA(Zhang et al., 2024) | 0.000 | 0.000 | 0.214 | 0.000 | 0.340 | 0.208 | 0.000 |
|   w/ Heat Map | 0.000 | 0.000 | 0.216 | 0.000 | 0.349 | 0.209 | 0.057 |
|   w/ Text | 0.000 | **0.036** | 0.214 | 0.000 | 0.346 | 0.209 | 0.040 |
|   w/ RadEyeVideo (Ours) | 0.000 | 0.000 | 0.215 | 0.000 | 0.346 | 0.208 | 0.050 |
| VideoLLaMA2(Cheng et al., 2024) | 0.000 | 0.000 | 0.217 | 0.000 | 0.352 | 0.209 | 0.031 |
|   w/ Heat Map | 0.000 | 0.031 | 0.202 | 0.000 | 0.329 | 0.205 | 0.057 |
|   w/ Text | 0.000 | 0.000 | 0.214 | 0.000 | 0.329 | 0.193 | 0.058 |
|   w/ RadEyeVideo (Ours) | 0.000 | 0.000 | 0.191 | 0.000 | 0.317 | 0.193 | 0.059 |
| LLaVA-OneVision(Li et al., 2024) | 0.000 | 0.000 | 0.217 | 0.000 | 0.338 | 0.207 | 0.032 |
|   w/ Heat Map | 0.000 | 0.000 | 0.212 | 0.000 | 0.338 | 0.204 | 0.072 |
|   w/ Text | 0.000 | 0.000 | **0.219** | 0.000 | 0.295 | 0.209 | 0.045 |
|   w/ RadEyeVideo (Ours) | 0.000 | 0.000 | 0.213 | 0.000 | **0.350** | 0.209 | 0.059 |

| Methods | Pneumonia | Atelectasis | Pneumothorax | Pleural Effusion | Pleural Other | Fracture | Support Devices |
|---|---|---|---|---|---|---|---|
| CheXagent(Chen et al., 2024b) | 0.190 | 0.000 | 0.000 | 0.000 | 0.000 | 0.000 | 0.000 |
| CXR-LLaVA(Lee et al., 2023) | 0.196 | 0.000 | 0.000 | 0.006 | 0.000 | 0.000 | 0.000 |
| LLaVA-Med(Li et al., 2024) | 0.195 | 0.000 | 0.000 | 0.000 | 0.000 | 0.000 | 0.000 |
| LongVA(Zhang et al., 2024) | 0.197 | 0.000 | 0.000 | 0.052 | 0.000 | 0.000 | 0.043 |
|   w/ Heat Map | 0.195 | 0.000 | 0.000 | 0.032 | 0.000 | 0.000 | 0.012 |
|   w/ Text | **0.198** | 0.000 | 0.000 | 0.227 | 0.000 | 0.000 | 0.000 |
|   w/ RadEyeVideo (Ours) | 0.196 | 0.000 | 0.000 | 0.141 | 0.000 | 0.000 | 0.011 |
| VideoLLaMA2(Cheng et al., 2024) | 0.196 | 0.070 | 0.037 | 0.197 | 0.000 | 0.000 | 0.183 |
|   w/ Heat Map | 0.195 | 0.113 | 0.038 | 0.249 | 0.000 | 0.000 | 0.107 |
|   w/ Text | 0.195 | **0.199** | **0.043** | **0.258** | 0.000 | 0.000 | 0.161 |
|   w/ RadEyeVideo (Ours) | 0.195 | 0.007 | 0.032 | 0.256 | 0.000 | 0.000 | 0.103 |
| LLaVA-OneVision(Li et al., 2024) | 0.199 | 0.128 | 0.000 | 0.089 | 0.000 | 0.000 | 0.077 |
|   w/ Heat Map | 0.195 | 0.085 | 0.038 | 0.043 | 0.000 | 0.000 | 0.077 |
|   w/ Text | 0.195 | 0.085 | 0.038 | 0.043 | 0.000 | 0.000 | 0.077 |
|   w/ RadEyeVideo (Ours) | 0.197 | 0.109 | 0.025 | 0.119 | 0.000 | 0.000 | **0.189** |

Table 10: Beta Split Diagnosis performance with CheXbert (F1 score for all abnormalities). All scores are raw scores reported. NoEye prompting scores are reported with the model name.

| Methods | No Finding | Enlarged Cardiomediastinum | Cardiomegaly | Lung Lesion | Lung Opacity | Edema | Consolidation |
|---|---|---|---|---|---|---|---|
| CheXagent(Chen et al., 2024b) | 0.000 | 0.000 | 0.000 | 0.000 | 0.125 | 0.400 | 0.000 |
| CXR-LLaVA(Lee et al., 2023) | 0.000 | 0.000 | 0.000 | 0.000 | 0.000 | 0.427 | 0.000 |
| LLaVA-Med(Li et al., 2024) | 0.000 | 0.000 | 0.000 | 0.000 | 0.000 | 0.427 | 0.000 |
| LongVA(Zhang et al., 2024) | 0.000 | 0.000 | 0.327 | 0.000 | 0.432 | 0.427 | 0.000 |
| w/ Heat Map | 0.000 | 0.000 | 0.386 | 0.000 | 0.496 | 0.427 | 0.081 |
| w/ Text | 0.000 | 0.000 | 0.382 | 0.000 | 0.496 | 0.435 | 0.000 |
| w/ RadEyeVideo (Ours) | 0.000 | 0.000 | **0.393** | 0.000 | **0.504** | **0.435** | **0.133** |
| VideoLLaMA2(Cheng et al., 2024) | 0.000 | 0.000 | 0.386 | 0.000 | 0.500 | 0.414 | 0.000 |
| w/ Heat Map | 0.000 | **0.125** | 0.333 | 0.000 | 0.495 | 0.467 | 0.112 |
| w/ Text | 0.000 | 0.000 | 0.369 | 0.000 | 0.472 | 0.424 | 0.106 |
| w/ RadEyeVideo (Ours) | 0.000 | 0.000 | 0.377 | 0.000 | 0.490 | 0.346 | 0.067 |
| LLaVA-OneVision(Li et al., 2024) | 0.000 | 0.000 | 0.386 | 0.000 | 0.451 | 0.427 | 0.000 |
| w/ Heat Map | 0.000 | 0.000 | 0.393 | 0.000 | 0.477 | 0.414 | 0.000 |
| w/ Text | 0.000 | 0.000 | 0.386 | 0.000 | 0.232 | 0.427 | 0.000 |
| w/ RadEyeVideo (Ours) | 0.000 | 0.000 | 0.384 | 0.000 | 0.492 | 0.427 | 0.000 |

| Methods | Pneumonia | Atelectasis | Pneumothorax | Pleural Effusion | Pleural Other | Fracture | Support Devices |
|---|---|---|---|---|---|---|---|
| CheXagent(Chen et al., 2024b) | 0.200 | 0.000 | 0.000 | 0.000 | 0.000 | 0.000 | 0.000 |
| CXR-LLaVA(Lee et al., 2023) | 0.196 | 0.000 | 0.000 | 0.000 | 0.000 | 0.000 | 0.000 |
| LLaVA-Med(Li et al., 2024) | 0.196 | 0.000 | 0.000 | 0.000 | 0.000 | 0.000 | 0.000 |
| LongVA(Zhang et al., 2024) | 0.206 | 0.000 | 0.000 | 0.077 | 0.000 | 0.000 | 0.000 |
| w/ Heat Map | 0.198 | 0.000 | 0.000 | 0.000 | 0.000 | 0.000 | 0.000 |
| w/ Text | 0.200 | 0.000 | 0.000 | 0.292 | 0.000 | 0.000 | 0.000 |
| w/ RadEyeVideo (Ours) | 0.206 | 0.000 | 0.000 | 0.114 | 0.000 | 0.000 | 0.000 |
| VideoLLaMA2(Cheng et al., 2024) | 0.198 | 0.000 | 0.000 | 0.305 | 0.000 | 0.000 | 0.259 |
| w/ Heat Map | 0.196 | 0.077 | 0.087 | 0.389 | 0.000 | 0.000 | 0.146 |
| w/ Text | 0.196 | 0.158 | 0.056 | 0.389 | 0.000 | 0.000 | 0.263 |
| w/ RadEyeVideo (Ours) | 0.196 | 0.000 | 0.056 | **0.389** | 0.000 | 0.000 | 0.174 |
| LLaVA-OneVision(Li et al., 2024) | 0.188 | **0.207** | 0.000 | 0.138 | 0.000 | 0.000 | **0.370** |
| w/ Heat Map | **0.217** | 0.000 | 0.000 | 0.154 | 0.000 | 0.000 | 0.077 |
| w/ Text | 0.118 | 0.077 | 0.000 | 0.000 | 0.000 | 0.000 | 0.091 |
| w/ RadEyeVideo (Ours) | 0.188 | 0.182 | 0.000 | 0.111 | 0.000 | 0.000 | 0.245 |

