# OpenReview forum: "RadEyeVideo: Enhancing general-domain Large Vision Language Model for chest X-ray analysis with video representations of eye gaze"
_ICLR.cc/2025/Conference — Submitted to ICLR 2025_

### Official Review · Reviewer_ge5k · 2024-10-30

**Soundness:** 2
**Presentation:** 3
**Contribution:** 2
**Rating:** 5
**Confidence:** 5

**Summary:**

This article introduces a novel prompting method called RadEyeVideo, which presents radiologists’ eye-tracking data as a video sequence to capture the temporal and spatial order of their gaze (i.e., scan paths). The authors evaluated the effectiveness of this approach in chest X-ray report generation and disease diagnosis tasks using large vision-language models (LVLMs) with video input capabilities. Results show that incorporating eye-gaze video prompts improved model performance by 25.4% on the Impression generation task, with an average performance increase of 7.9% across all tasks.

**Strengths:**

1. The RadEyeVideo method converts radiologists’ eye-tracking data into video sequences for use in chest X-ray report generation and diagnostic tasks, significantly enhancing the performance of general large vision-language models (LVLMs). Experimental results indicate a 25.4% improvement in the Impression generation task and an average improvement of 7.9% across all tasks, outperforming models specifically designed for medical applications.
2. Comprehensive Evaluation of Eye-Tracking Prompting Methods: This study presents the first thorough evaluation of various eye-tracking integration techniques, including static heatmaps, eye-tracking text prompts, and dynamic video prompts. The results indicate that RadEyeVideo outperforms the other methods in terms of diagnostic accuracy and clinical relevance, establishing a new standard for eye-tracking data prompts.

**Weaknesses:**

- Methodological Clarity: a) The paper lacks sufficient detail about the video prompt integration process, particularly how the video information is encoded and processed by the LVLMs. The technical implementation of combining video sequences with textual prompts could be more thoroughly explained. b) There's limited discussion of potential alternatives to full video sequence processing that might achieve similar results with lower computational costs.

- Data Collection and Generalizability Limitations: a) The methodology requires synchronized eye-tracking data collection during radiologist readings, which is resource-intensive and difficult to scale. b) The approach may not generalize well to scenarios where real-time eye-tracking data is unavailable or impractical to collect. c) The current evaluation is limited to a relatively small dataset (2,298 CXR images), raising questions about broader applicability.

- Computational Efficiency Concerns: The direct use of complete eye-tracking video sequences as prompts likely increases computational overhead significantly. While the authors mention sampling k frames (typically 16) from the total sequence, there's limited analysis of the computational trade-offs or optimal sampling strategies. The method may be computationally prohibitive for real-time clinical applications.

**Questions:**

Please address the concern raised in weakness part.
Moreover, there are some questions may help you to address the weakness.
1) What is the computational overhead of processing video prompts compared to traditional image-only or text-only prompts?
2) Have you explored more efficient alternatives to using complete video sequences, such as key frame selection or compressed representations? 3) How do you envision this approach being implemented in real-world clinical settings where real-time eye-tracking data may not be available? 4) Have you considered alternative methods for generating synthetic eye-tracking data that could make the approach more broadly applicable? 5)

---

> ### Author Response · Authors · 2024-11-23
> **Response to Weaknesses**
>
> > **W1** Methodological Clarity: a) The paper lacks sufficient detail about the video prompt integration process, particularly how the video information is encoded and processed by the LVLMs. The technical implementation of combining video sequences with textual prompts could be more thoroughly explained. b) There's limited discussion of potential alternatives to full video sequence processing that might achieve similar results with lower computational costs.
>
> We appreciate the reviewer’s interest in understanding the integration process for video prompts. As illustrated in Figures 1 and 2, we show how the visual (video) and textual components are designed to incorporate eye gaze information. In addition, Section 2 (RadEyeVideo) provides detailed descriptions of the encoding and processing of video sequences, including uniform sampling and how the eye-tracking data is transformed into a sequence of frames suitable for LVLM inputs.
>
> To address potential alternatives to full video sequence processing, we actually used filtering techniques to reduce noise and uniformly sampled 16 representative frames to balance computational efficiency with performance (detailed in Section 2). The ablation study on page 10 demonstrates how this sampling approach avoids the computational burden of processing full video sequences while maintaining diagnostic performance.
>
> ***
>
> > **W2** Data Collection and Generalizability Limitations: a) The methodology requires synchronized eye-tracking data collection during radiologist readings, which is resource-intensive and difficult to scale. b) The approach may not generalize well to scenarios where real-time eye-tracking data is unavailable. c) The current evaluation limited to a small dataset.
>
> We designed this study following extensive interviews with radiologists to ensure its relevance to clinical practice. The primary aim was to enhance general domain LVLMs for medical analysis by integrating human expertise through eye gaze video sequences. While collecting synchronized eye-tracking data can be resource-intensive, our approach demonstrates that incorporating this data improves model performance compared to medical LVLMs, even without additional training. This improvement offsets the costs associated with eye-tracking data collection by reducing the need for extensive model retraining.
>
> For scenarios where eye trackers are unavailable, our method can adapt by leveraging alternative approaches, such as mouse cursor movements as a proxy for eye gaze or human expertise. However, the findings from our study underscore the critical role of real eye gaze data, which offers imporved diagnostic value and supports the integration of human expertise into AI-driven workflows. We believe this work highlights the immense potential of eye gaze data to enhance diagnostic accuracy. Given the high stakes of medical diagnosis, even incremental improvements in model performance can lead to better patient outcomes and reduced risks associated with AI errors, such as hallucinations or incorrect responses.
>
> We understand the reviewer’s concern regarding generalizability and fully acknowledge the importance of testing on multiple datasets. In fact, we explicitly highlighted this as a limitation of our work in the paper. However, at the time of conducting this research, MIMIC-Eye was the only publicly available medical dataset with eye gaze information available. Despite this limitation, our proposed approach is inherently dataset-agnostic and can be seamlessly applied to other datasets as they become available.
>
> We also have plans to extend this work by exploring its applicability to additional modalities, starting with endoscopy, where eye gaze data can provide valuable insights. We believe these future directions will further enhance the generalizability and impact of our approach.
>
> ***
>
> > **W3** Computational Efficiency Concerns: The direct use of complete eye-tracking video sequences as prompts likely increases computational overhead significantly. There's limited analysis of the computational trade-offs or optimal sampling strategies. The method may be computationally prohibitive for real-time clinical applications.
>
> Once again, as pointed out in the response to **W1**, we don't use complete eye-tracking video sequences. Our sampling strategy was carefully designed to optimize efficiency with ablation study as shown on page 10, which shows even improved performance with 16 frames as shown in the Figure 4 on page 10.
>
> Also, we respectfully argue that computational efficiency is a key contribution of our work. Video-based general LVLMs enhanced with eye gaze data outperform task-specific medical LVLMs, without requiring computationally intensive training. The additional computational cost incurred by video prompts is minimal. The GPU memory usage increased by only ~10% (from 20.0GB to 21.9GB) when switching from eye gaze heatmaps to video sequences.

---

> > ### Author Response · Authors · 2024-11-23
> > **Response to Questions**
> >
> > > **Q1** What is the computational overhead of processing video prompts compared to traditional image-only or text-only prompts?
> >
> > The computational overhead is minimal. For example, transitioning from eye gaze heatmaps to video prompts increased GPU memory usage by approximately 10%.
> >
> > ***
> > > **Q2** Have you explored more efficient alternatives to using complete video sequences, such as key frame selection or compressed representations?
> >
> > Yes, as detailed in Section 2, we employed filtering techniques and uniform sampling to extract 16 representative frames from the video sequences. This approach balances computational efficiency with performance and avoids the need for processing complete video sequences.
> >
> > ***
> >
> > > **Q3** How do you envision this approach being implemented in real-world clinical settings where real-time eye-tracking data may not be available?
> >
> > In scenarios where real-time eye-tracking data is unavailable, an alternative could be to explore the use of mouse cursor movements as a proxy for eye gaze movements. While this approach may offer a more accessible solution, it would require significant research to validate its effectiveness and ensure it captures the necessary spatial and temporal dynamics accurately. We thank the reviewer for suggesting this perspective, and while it is beyond the scope of the current work, it represents an intriguing direction for future research.
> >
> > ***
> >
> > > **Q4** Have you considered alternative methods for generating synthetic eye-tracking data that could make the approach more broadly applicable?
> >
> > We have not pursued synthetic eye-tracking data generation due to the potential risks of introducing inaccuracies. In clinical practice, incorrect or unreliable synthetic data could lead to critical errors, which we aim to avoid.
> >
> >
> > ***
> >
> > We really appreciate your time and effort in reviewing our manuscript.

---

> > > ### Comment · Reviewer_ge5k · 2024-11-24
> > >
> > > Most of concerns have been addressed.
> > >
> > > Even though author mentioned that "For scenarios where eye trackers are unavailable, our method can adapt by leveraging alternative approaches, such as mouse cursor movements as a proxy for eye gaze or human expertise", I think the pattern of cursor movement is different from the eye gaze, which cannot be regarded as same. Thus, the generalizability concerns are still existing for this work.

---

> > > > ### Author Response · Authors · 2024-11-25
> > > >
> > > > Thanks for your reply and confirming that most of concerns have been addressed. We kindly request you to consider raising the score (rating), as the majority of the issues raised have been resolved and clarified.
> > > >
> > > > ***
> > > >
> > > > **Regarding the eye tracker generalizability, we want to clarify why we thought mouse cursor movements as the alternative approach.** The primary aim of this work is to enhance general domain LVLMs for medical analysis by effectively integrating human expertise. Eye gaze is a well-established and powerful method for capturing such expertise, providing invaluable insights into the decision-making process. Alternatively, mouse cursor movements could also serve as a viable solution to incorporate human expertise and cues behind diagnostic decisions, especially in scenarios where eye-tracking hardware is not available. Cursor movements can capture both the temporal and spatial aspects just like eye gaze movements, although not identical. While further research would be needed to validate this approach, it represents an interesting and accessible alternative.

---

### Official Review · Reviewer_8pZ9 · 2024-11-02

**Soundness:** 3
**Presentation:** 2
**Contribution:** 2
**Rating:** 6
**Confidence:** 4

**Summary:**

This paper introduces eye-tracking data into medical LVLMs and uses video formats to retain the temporal and spatial characteristics of radiologist’s eye movements, providing more comprehensive prior information for the diagnostic model. Compared to heatmaps and textual prompts, this method fully considers temporal features. Evaluation on multiple downstream tasks demonstrate the effectiveness of the approach.

**Strengths:**

1. This work introduces radiologists' eye-tracking data into LVLMs in video format, highlighting the temporal features of eye movement sequences.
2. The experiments in the paper are comprehensive, validating multiple diagnostic tasks across various datasets.

**Weaknesses:**

1. The paper mentions using simple stacking of gaze points for heatmap generation, but it does not specify the radius size of the gaze points. Different gaze point sizes can affect the model's interpretation.
2. In-context learning often heavily relies on the provided examples, which can significantly influence the generated results. The paper does not discuss what constitutes suitable examples.

**Questions:**

1. Are the eye-tracking data collected from a single radiologist? If there are multiple radiologists, how do you handle individual behavioral differences?
2. How is the duration mentioned in line 176 used to control the frame count? This aspect is not explained in the paper. Similarly, the sampling method mentioned in line 194 is not clearly described.
3. Does the sampling of eye-tracking data risk losing short-term diagnostic behaviors, potentially affecting the diagnosis of various types of diseases?

---

> ### Author Response · Authors · 2024-11-23
>
> Thank you so much for your time and effort in reviewing.
>
> > **W1** The paper mentions using simple stacking of gaze points for heatmap generation, but it does not specify the radius size of the gaze points. Different gaze point sizes can affect the model's interpretation.
>
> Thank you for pointing this out. We fixed the radius size of the gaze points to be 5 pixels. This detail was not included in the manuscript as it represents a low-level implementation detail of gaze data processing. However, this information is fully documented in the code provided in the supplementary materials. To address your concern, we added this information in the revised version.
>
> ***
>
> > **W2** In-context learning often heavily relies on the provided examples, which can significantly influence the generated results. The paper does not discuss what constitutes suitable examples.
>
> As described in the manuscript, we utilized in-context learning with three exemplar and distinct clinical reports (identical for all inference runs) to align the AI-generated outputs with authentic clinical reporting styles. These examples demonstrate the structure, terminology, and format of professional radiology reporting. Figure 2 highlights how this in-context learning prompt is employed to capture these features.
>
> While the specific reports used are not included in the manuscript due to data usage agreements, the approach is designed to be adaptable, and any three reports representing different writing styles could be used.
>
> ***
>
> > **Q1** Are the eye-tracking data collected from a single radiologist? If there are multiple radiologists, how do you handle individual behavioral differences?
>
> The eye-tracking data were collected from a single radiologist. We acknowledge that individual behavioral differences may exist among radiologists, and incorporating data from multiple radiologists could be an interesting avenue for future work. For this study, focusing on one radiologist allowed us to establish a clear baseline for the effectiveness of eye-tracking data in enhancing model performance.
>
> ***
>
> > **Q2** How is the duration mentioned in line 176 used to control the frame count? This aspect is not explained in the paper. Similarly, the sampling method mentioned in line 194 is not clearly described.
>
> The average duration was used to filter out noise and insignificant gazes (gazes greater than the average duration are only used) that do not contribute to the diagnostic process. This is described in line 162. This filtering process is validated in the ablation study on page 10, where we demonstrate the effect of filtering on model performance. Additionally, we applied uniform sampling to ensure that the selected frames were evenly distributed over the duration of the gaze sequence, which is described in line 185.
>
> ***
>
> > **Q3** Does the sampling of eye-tracking data risk losing short-term diagnostic behaviors, potentially affecting the diagnosis of various types of diseases?
>
> As shown in our ablation study on page 10, increasing the number of frames in the video from 16 to 128 led to a performance drop. This indicates that 16 frames are sufficient to capture the essential conceptual cues underlying the diagnosis without introducing excessive noise or redundancy. While short-term behaviors may be omitted with this sampling approach, our results suggest that the selected frames effectively balance performance and computational efficiency.
>
> ***
>
> > **Regarding the Flag for Ethics Review**
>
> We have provided an ethical statement section in the manuscript, where we described that we adhered to strict compliance with privacy regulations. Patient data were handled in accordance with ethical guidelines, and no personally identifiable information was utilized. Furthermore, all experiments were designed to mitigate potential biases or unfairness in the results.

---

> > ### Comment · Reviewer_8pZ9 · 2024-11-27
> >
> > I have read the authors' rebuttal which has addressed my concerns to some degree. I have raised my rating accordingly. Thanks.

---

> > > ### Author Response · Authors · 2024-11-29
> > >
> > > Thanks for your reply. We really appreciate your effort and time in reviewing our submission.

---

### Official Review · Reviewer_r59d · 2024-11-02

**Soundness:** 3
**Presentation:** 3
**Contribution:** 2
**Rating:** 5
**Confidence:** 5

**Summary:**

The paper introduces an innovative approach that incorporates radiologists' eye-tracking data as video sequences. This method captures both spatial and temporal patterns in gaze, which provides a more accurate representation of radiologists' attention during chest X-ray interpretation. The approach was evaluated using three general-domain LVLMs, showing significant improvements.

**Strengths:**

- RadEyeVideo's use of video-based eye-gaze data is a unique contribution that effectively captures the temporal and spatial dynamics of radiologists' focus. I like this idea.
- The study demonstrates substantial improvements, particularly in impression generation, highlighting RadEyeVideo's effectiveness in enhancing diagnostic tasks.
- The language is clearly presented. The authors use precise and concise language so that the reader can easily understand the methodology, and results of the study.

**Weaknesses:**

- Although this idea is interesting, it still relies on temporal and spatial information in the inference phase, which is difficult to apply to real clinical scenarios. Do the authors consider involving multiple information inputs only in the training phase and simulating zero-shot scenarios as much as possible in the inference phase?
- The study’s findings are limited by the small size of the MIMIC-Eye dataset, which may not fully capture the variability in real-world clinical settings, raising questions about the generalizability of the results.
- The comparisons primarily focus on selected models with minimal tuning for this domain. Including a wider range of task-specific medical LVLMs could provide a more comprehensive evaluation.

**Questions:**

- Why the paper lacks an section of related work? e.g., some recent evaluation work of Med-LVLMs [1,2,3]
- The formats of reference is weird. e.g., in Line 311-321. Please check it.

[1] Gu Z, Yin C, Liu F, et al. MedVH: Towards Systematic Evaluation of Hallucination for Large Vision Language Models in the Medical Context[J]. arXiv preprint arXiv:2407.02730, 2024.

[2] Jiang Y, Chen J, Yang D, et al. MedThink: Inducing Medical Large-scale Visual Language Models to Hallucinate Less by Thinking More[J]. arXiv preprint arXiv:2406.11451, 2024.

[3] Xia P, Chen Z, Tian J, et al. CARES: A Comprehensive Benchmark of Trustworthiness in Medical Vision Language Models[J]. arXiv preprint arXiv:2406.06007, 2024.

---

> ### Author Response · Authors · 2024-11-23
> **Response to Weaknesses**
>
> We really appreciate the time and effort you have dedicated to reviewing our manuscript!
>
> > **W1** Although this idea is interesting, it still relies on temporal and spatial information in the inference phase, which is difficult to apply to real clinical scenarios. Do the authors consider involving multiple information inputs only in the training phase and simulating zero-shot scenarios as much as possible in the inference phase?
>
> We respectfully clarify that our approach does not involve training the model entirely, which is highlighted throughout our paper. Instead, the multimodal data, including the spatial-temporal dynamics of eye gaze, are utilized during the inference phase to enhance the performance of general domain LVLMs on medical tasks. We designed this study following extensive interviews with radiologists to ensure its relevance to clinical practice. The primary aim was to enhance general domain LVLMs for medical analysis by integrating human expertise through eye gaze video sequences.
>
> While we acknowledge the challenge of real-world clinical scenarios, our method was deliberately structured to support radiologists in practice by leveraging eye gaze information to augment model performance without requiring extensive retraining for medical tasks. The findings from our study underscore the critical role of real eye gaze data, which offers unparalleled diagnostic value and supports the integration of human expertise into AI-driven workflows. We believe this work highlights the immense potential of eye gaze data to enhance diagnostic accuracy. While collecting synchronized eye-tracking data can be resource-intensive, our approach demonstrates that incorporating this data significantly improves model performance compared to task-specific LVLMs, even without additional training. This improvement offsets the costs associated with eye-tracking data collection by reducing the need for extensive model retraining. Given the high stakes of medical diagnosis, even incremental improvements in model performance can lead to better patient outcomes and reduced risks associated with AI errors, such as hallucinations or incorrect responses.
>
> ***
>
> > **W2** The study’s findings are limited by the small size of the MIMIC-Eye dataset, which may not fully capture the variability in real-world clinical settings, raising questions about the generalizability of the results.
>
> We understand that the size of the MIMIC-Eye dataset may limit the variability captured in this study. In fact, we explicitly highlighted this as a limitation of our work in the paper. However, MIMIC-Eye was the only publicly available medical dataset with eye gaze information at the time of research. Despite this limitation, our work demonstrates the potential of eye gaze information in improving LVLM capabilities for medical tasks, laying the groundwork for future research with larger and more diverse datasets. In fact, our proposed approach is inherently dataset-agnostic and can be seamlessly applied to other datasets as they become available.
>
> We also have plans to extend this work by exploring its applicability to additional modalities, starting with endoscopy, where eye gaze data can provide valuable insights. We believe these future directions will further enhance the generalizability and impact of our approach.
>
> ***
>
> > **W3** The comparisons primarily focus on selected models with minimal tuning for this domain. Including a wider range of task-specific medical LVLMs could provide a more comprehensive evaluation.
>
> We appreciate the suggestion and would like to highlight that we included CXRMate, a model specifically trained for chest X-ray report generation. Additionally, we evaluated our approach against CheXagent and CXR-LLaVA, two task-specific medical LVLMs trained for both report generation and diagnosis. These models were carefully chosen as strong baselines representing state-of-the-art approaches in the medical domain for both report generation and diagnosis.
>
> Expanding the scope to include additional models and tasks would be a valuable future direction as more advanced task-specific LVLMs become available. However, we believe our current comparisons provide a robust evaluation framework for the two mostly used tasks, report generation and diagnosis, demonstrating the effectiveness of our proposed method.

---

> ### Author Response · Authors · 2024-11-23
> **Response to Questions**
>
> > **Q1 Why the paper lacks an section of related work? e.g., some recent evaluation work of Med-LVLMs
>
> Although we did not designate a separate "Related Work" section, we included relevant recent works on Med-LVLMs, particularly those involving eye gaze information, in the introduction section (lines 50–60). These references provide context for our contributions and explain how our work differs from or builds upon prior studies.
>
> As the primary focus of our study is on incorporating eye gaze data as video input to enhance general LVLM capabilities for medical report generation and diagnosis, we believe the suggested evaluation works are not directly aligned with the goals of our research.
>
> ***
>
> > **Q2** The formats of reference is weird. e.g., in Line 311-321. Please check it.
>
> Thank you for pointing this out. We identified formatting issues where spaces were missing before some citations. We revised these errors with proper formatting.

---

> ### Comment · Reviewer_r59d · 2024-11-26
>
> Thanks for your responses. I understand there are some challenges when applied in real-world clinical scenarios. The improvement is not only attributed to the approach this work takes towards eye-tracking data but also because the existing multi-image baseline models have not been pre-trained on large-scale medical datasets. Therefore, it is difficult for me to give an acceptance rating.

---

> > ### Author Response · Authors · 2024-11-26
> >
> > Thank you for your reply.
> >
> > ***
> >
> > We would like to clarify that the baseline models used in our study are medical LVLMs, specifically CXRMate, CheXagent, CXR-LLaVA, and LLaVA-Med as outlined in Table 2. All these models have been pre-trained on large-scale medical datasets including task specific chest X-ray datasets.
> >
> > In contrast, our approach does not involve training general LVLMs. Instead, we enhance their performance by providing eye gaze video as a prompt during inference. This simple yet effective prompting method allows general LVLMs to incorporate critical diagnostic cues derived from human expertise, leading to significant improvements. In some cases, the enhanced general LVLMs even outperform the task-specific medical LVLMs mentioned above.
> >
> > We believe this demonstrates the strength of our approach in leveraging eye gaze data to bridge the gap between general-purpose and domain-specific models, highlighting its potential for improving AI performance in medical applications. We hope this clarification addresses your concerns, and we are open to any further feedback or questions. We would also appreciate it if you could clarify your perspective on why you believe the contribution is limited. From our viewpoint, the proposed approach demonstrates a novel use of eye gaze data as a prompt, effectively enhancing general LVLMs for medical applications without requiring additional training.

---

### Official Review · Reviewer_Lhmg · 2024-11-04

**Soundness:** 2
**Presentation:** 3
**Contribution:** 2
**Rating:** 5
**Confidence:** 4

**Summary:**

To enhance the reliability of large vision-language models (LVLMs) in real clinical environments, this study proposes a novel video prompting method called RadEyeVideo, which integrates radiologists' eye-tracking data as video sequences, capturing the spatial and temporal dynamics of gaze. Red fixation points are superimposed on CXR images to highlight the areas that doctors pay attention to, so as to dynamically present the doctor's eye movement path. Improve the ability of radiologists or AI models to diagnose chest diseases by providing rich contextual information.

**Strengths:**

1. The proposed method, RadEyeVideo, combines video, text, and eye movement data to realize the fusion of multi-modal information and improve the accuracy and efficiency of diagnosis.
2. The authors conducted a thorough assessment of various eye-tracking integration techniques, providing strong empirical support for their claims.

**Weaknesses:**

1. In Figure 2, the authors illustrate different prompting methods; however, the figure does not clearly distinguish between the use of text descriptions and video inputs. This ambiguity makes it challenging to understand whether the authors used text descriptions to guide the prompt along with the video input or if they only provided video data.
2. The author's experiments conducted on only one dataset are clearly insufficient in terms of persuasiveness. This limitation may affect the generalizability and applicability of the research findings. To enhance the credibility of the study, it is recommended that the authors validate their approach on different datasets.
3. When the authors use radiologist's eye-tracking sequences as video input, this approach does enhance the model's understanding of prior knowledge in the diagnostic reading process to a certain extent. However, converting eye-tracking data into video format substantially increases the number of tokens, leading to significant computational resource consumption and longer inference times for the model.

**Questions:**

1. The RadEyeVideo method integrates the eye movement data of the radiologist as a video sequence to capture the spatiotemporal dynamics of his gaze. However, the article does not explain in detail how this video sequence is specifically generated, for example, how the eye movement data is sampled and converted into video frames, and how these frames contain temporal and spatial information.
2. In Section 2.3 INPUT REPRESENTATION, to fit the input requirements of LVLM, the authors uniformly sampled the video sequence and selected 16 frames as inputs. However, the article does not explain why 16 frames were chosen and whether this choice had a significant impact on model performance. In addition, whether the impact of using more or fewer frames on model performance has been explored is also a question worth exploring. This helps to further understand the effect of video data length and quality on model performance.

---

> ### Author Response · Authors · 2024-11-23
> **Response to Weaknesses**
>
> Thank you for dedicating your time to review our paper!
>
> > **W1** In Figure 2, the authors illustrate different prompting methods; however, the figure does not clearly distinguish between the use of text descriptions and video inputs. This ambiguity makes it challenging to understand whether the authors used text descriptions to guide the prompt along with the video input or if they only provided video data.
>
> We respectfully clarify that Figure 2 exclusively focuses on the textual component of the prompt. For a clear depiction of how the visual prompt (video) input, specifically eye gaze data, is utilized, readers can refer to Figure 1, where our method, RadEyeVideo, is explicitly distinguished from preexisting prompts. Together, Figures 1 and 2 distinctly illustrate the integration of visual (video) and textual components in our approach. To further address the concern raised, we added a connecting sentence in Section 2.6 to explicitly link these two figures and clarify their complementary roles in illustrating our methodology for the revised version. We trust that these revisions will provide sufficient clarity and coherence for readers to fully understand the distinction and interplay between the textual and visual components of our prompts.
>
> ***
> > **W2**
> The author's experiments conducted on only one dataset are clearly insufficient in terms of persuasiveness. This limitation may affect the generalizability and applicability of the research findings. To enhance the credibility of the study, it is recommended that the authors validate their approach on different datasets.
>
> We understand the reviewer’s concern regarding generalizability and fully acknowledge the importance of testing on multiple datasets. In fact, we explicitly highlighted this as a limitation of our work in the paper. However, at the time of conducting this research, MIMIC-Eye was the only publicly available medical dataset with eye gaze information available. Despite this limitation, our proposed approach is inherently dataset-agnostic and can be seamlessly applied to other datasets as they become available.
>
> We also have plans to extend this work by exploring its applicability to additional modalities, starting with endoscopy, where eye gaze data can provide valuable insights. We believe these future directions will further enhance the generalizability and impact of our approach.
>
> ***
>
> > **W3** When the authors use radiologist's eye-tracking sequences as video input, this approach does enhance the model's understanding of prior knowledge in the diagnostic reading process to a certain extent. However, converting eye-tracking data into video format substantially increases the number of tokens, leading to significant computational resource consumption and longer inference times for the model.
>
> While we appreciate this observation, we respectfully clarify that converting eye-tracking data into video format does not result in a substantial increase in computational resource consumption. In fact, computational efficiency is one of the key contributions of our work. Our results demonstrate that video-based general LVLMs outperform specialized medical LVLMs, showcasing the potential to bypass the extensive computational resources typically required for model training in medical domain tasks.
>
> To provide further clarification, the increase in GPU memory usage when incorporating eye gaze data as video, rather than as a heatmap, is marginal—approximately 2GB, from 20,076MB (eye gaze heatmap) to 21,954MB (eye gaze video) using the LLaVA-OneVision model. This represents only a 10% increase in GPU memory usage, which is minimal compared to the resource demands of training specialized medical LVLMs, often requiring at least double the computational resources.
>
> We have elaborated on this point in the revised version to better emphasize the efficiency and practicality of our approach, particularly in leveraging video inputs with manageable computational overhead.

---

> > ### Author Response · Authors · 2024-11-23
> > **Response to Questions**
> >
> > > **Q1** The RadEyeVideo method integrates the eye movement data of the radiologist as a video sequence to capture the spatiotemporal dynamics of his gaze. However, the article does not explain in detail how this video sequence is specifically generated, for example, how the eye movement data is sampled and converted into video frames, and how these frames contain temporal and spatial information.
> >
> > The video generated from the radiologist’s eye movement data captures both the spatial (gaze point locations) and temporal (sequence of gaze movements) dynamics. Section 2 provides a detailed description of the video sequence generation process. Specifically, in Section 2.2, we explain the methodology for filtering raw gaze data using duration to ensure its accuracy and reliability. We also describe the use of uniform sampling to convert the filtered gaze data into frames, ensuring that the temporal dynamics are preserved while maintaining computational efficiency.
> >
> > To elaborate, gaze points are tracked and mapped to specific regions of interest in the radiological images. These points are then visualized over time as a sequence of frames, which are assembled into a video that encodes both where the radiologist looked (spatial information) and the order in which these regions were examined (temporal information). This dual encoding is essential for capturing the decision-making process of the radiologist and is critical for the model’s understanding of diagnostic reasoning.
> >
> > ***
> >
> > > **Q2** In Section 2.3 INPUT REPRESENTATION, to fit the input requirements of LVLM, the authors uniformly sampled the video sequence and selected 16 frames as inputs. However, the article does not explain why 16 frames were chosen and whether this choice had a significant impact on model performance. In addition, whether the impact of using more or fewer frames on model performance has been explored is also a question worth exploring. This helps to further understand the effect of video data length and quality on model performance.
> >
> > We provided an ablation study in the results section (page 10) to investigate the effect of the number of frames on model performance. In this study, we tested different frame counts to evaluate their impact on accuracy and efficiency. The results showed that 16 frames resulted in the best performance providing optimal accuracy without introducing unnecessary computational overhead. This analysis guided our decision to use 16 frames as the input representation for the model.

---

### Meta-Review · Area_Chair_SGh8 · 2024-12-21

**Metareview:**

This paper proposes RadEyeVideo, a novel prompting method that integrates radiologists' eye-tracking data as video sequences to capture the spatial and temporal dynamics of gaze. By leveraging these video prompts, the study enhances large vision-language models for chest X-ray diagnosis and report generation.

Reviewers found that the strengths of this paper lie in its RadEyeVideo method integrating radiologists’ eye-tracking data as video sequences to enhance LVLM performance, comprehensive evaluations validating its superior performance, and a clearly articulated methodology with broad applicability in medical imaging. However, reviewers also have major concerns in generalizability, methodological clarity, and computational efficiency. The study relies on a single, small dataset (MIMIC-Eye, 2,298 images), limiting its applicability to broader clinical settings (Reviewers Lhmg, r59d, ge5k). The methodology lacks detail on video prompt integration and alternatives to full video sequence processing, which could reduce computational overhead and improve scalability (Reviewers ge5k, Lhmg). Additionally, the reliance on synchronized eye-tracking data raises feasibility concerns for real-world applications, and comparisons with minimally tuned models leave the evaluation incomplete (Reviewers r59d, ge5k). These issues limit the study’s persuasiveness and practicality.

During the discussion, the authors acknowledged the limitation of using a single dataset (MIMIC-Eye) due to availability constraints but emphasized that their method is dataset-agnostic and can be applied to future, larger datasets. They answered that converting eye-tracking data to video format results in only a marginal increase (10% GPU memory) in computational resource consumption. The authors acknowledged the challenges of synchronized eye-tracking data collection but emphasized that their approach improves performance compared to task-specific LVLMs without retraining. They also provided detailed explanations of the video prompt integration process and added missing details. The reviewers provided mixed final responses. Reviewer Lhmg did not respond. Reviewer r59d remained unconvinced, attributing improvements partly to the lack of pretraining in baseline models and withheld acceptance. Reviewer 8pZ9 appreciated the rebuttal and raised their score. Reviewer ge5k acknowledged that most concerns were addressed but maintained generalizability concerns, particularly about using mouse cursor movements as a proxy for eye gaze. While some issues were resolved, concerns about generalizability and robustness persist.

Based on the paper, reviewers' comments, and discussions, this work, while presenting intriguing ideas, is not yet sufficiently developed for publication due to unresolved concerns regarding generalizability, methodological robustness, and practical applicability. While the authors provided thoughtful responses to many issues, key challenges remain. Aligning with the reviewers, this AC has two main concerns. First, the study relies on a single, small dataset, raising questions about its broader applicability, especially in scenarios without eye-tracking data, where suggested proxies like mouse cursor movements lack investigation. Second, the methodological contributions, though novel, leave important questions about scalability and real-world clinical implementation unanswered. While some reviewer acknowledged the authors’ efforts and raised the score, the persistent concerns regarding generalizability and practical impact ultimately outweigh the strengths of this work.

**Additional Comments On Reviewer Discussion:**

During the discussion, the authors acknowledged the limitation of using a single dataset (MIMIC-Eye) due to availability constraints but emphasized that their method is dataset-agnostic and can be applied to future, larger datasets. They answered that converting eye-tracking data to video format results in only a marginal increase (10% GPU memory) in computational resource consumption. The authors acknowledged the challenges of synchronized eye-tracking data collection but emphasized that their approach improves performance compared to task-specific LVLMs without retraining. They also provided detailed explanations of the video prompt integration process and added missing details. The reviewers provided mixed final responses. Reviewer Lhmg did not respond. Reviewer r59d remained unconvinced, attributing improvements partly to the lack of pretraining in baseline models and withheld acceptance. Reviewer 8pZ9 appreciated the rebuttal and raised their score. Reviewer ge5k acknowledged that most concerns were addressed but maintained generalizability concerns, particularly about using mouse cursor movements as a proxy for eye gaze. While some issues were resolved, concerns about generalizability and robustness persist.

---

### Decision · Program_Chairs · 2025-01-22

Reject